# Patisiran in hATTR Amyloidosis: Six-Month Latency Period before Efficacy

**DOI:** 10.3390/brainsci11040515

**Published:** 2021-04-19

**Authors:** Luca Gentile, Massimo Russo, Marco Luigetti, Giulia Bisogni, Andrea Di Paolantonio, Angela Romano, Valeria Guglielmino, Ilenia Arimatea, Mario Sabatelli, Antonio Toscano, Giuseppe Vita, Anna Mazzeo

**Affiliations:** 1Unit of Neurology and Neuromuscular Diseases, Department of Clinical and Experimental Medicine, University of Messina, 98125 Messina, Italy; russom@unime.it (M.R.); ilenia.arimatea@hotmail.it (I.A.); atoscano@unime.it (A.T.); giuseppe.vitag@unime.it (G.V.); annamazzeo@yahoo.it (A.M.); 2UOC Neurologia, Fondazione Policlinico Universitario A. Gemelli IRCCS, 00168 Roma, Italy; mluigetti@gmail.com (M.L.); andrea.dp1988@gmail.com (A.D.P.); angela.romano12@gmail.com (A.R.); guglielmino.valeria@gmail.com (V.G.); mario.sabatelli@unicatt.it (M.S.); 3Dipartimento di Neuroscienze, Università, Cattolica del Sacro, Cuore, 00168 Roma, Italy; 4Centro Clinico NEMO, Fondazione Policlinico Universitario A. Gemelli IRCCS, 00168 Roma, Italy; giulia.bisogni@centrocliniconemo.it

**Keywords:** hATTR, amyloidosis, TTR silencers, gene therapy

## Abstract

Hereditary amyloidosis associated with mutations in the transthyretin gene (hATTR) is a progressive devastating disease, with a fatal outcome occurring within 10years after onset. In recent years, TTR gene silencing therapy appeared as a promising therapeutic strategy, showing evidence that disease progression can be slowed and perhaps reversed. We report here 18 subjects affected by hATTR amyloidosis treated with patisiran, a small interfering RNA acting as TTR silencer, and evaluated with a PND score, the NIS and NIS-LL scale, and a Norfolk QOL-DN questionnaire at baseline and then every 6 months. A global clinical stabilizationwas observed for the majority of the patients, with mild-moderate improvements in some cases, even in advanced disease stage (PND score > 2). Analysis of NIS, NIS-LL and Norfolk QOL-DN results, and PND score variation suggest the possible presence of a 6-month latency period prior to benefit of treatment.

## 1. Introduction

Hereditary amyloidosis associated with mutations in the transthyretin (TTR) gene (hATTR) is the most common form of genetic amyloidosis. It is a progressive devastating disease transmitted as an autosomal dominant trait, with a fatal outcome occurring within 10years after onset [1,2]. More than 130 TTR gene mutations have been identified thus far, with Val30Met as the most common mutation reported worldwide [3,4,5]. The liver is the primary source of circulating tetrameric TTR, but it is also synthesized by the retinal pigment epithelium and choroid plexus. TTR functions primarily as a transport protein for vitamin A and for approximately 15% of circulating plasma thyroxine. In patients with hATTR amyloidosis, the tetramer dissociates in dimers and, then, in monomers, which deposit in target tissues forming abnormal amyloid fibril aggregates [6]. However, in hATTR amyloidosis, both mutant and wild-type TTR deposit as amyloid in peripheral nerves and in many other organs, including heart, kidney, and gastrointestinal tract [6,7,8]. While the exact clinical phenotype depends on the underlying mutation, cardiomyopathy, peripheral polyneuropathy, and autonomic neuropathy with orthostatic hypotension and gastrointestinal dysautonomia are common [9], and the previously used definition of Familial Amiloidotic Cardiomyopathy (FAC) or Familial AmiloidoticPolineuropathy (FAP) have become outdated [10,11]. The necessity to monitor disease progression has recently led to an increase of the research on new biomarkers [12,13,14,15,16,17,18], with the particular aim to record the shift from asymptomatic to symptomatic stage, whichcould allow physicians to start specific treatment for hATTR amyloidosis.

Liver transplantation (LT) and combined heart–liver transplantation represented the first specific therapy for hATTR amyloidosis, suppressing the main source of mutant TTR. However, its effectiveness has been demonstrated mainly in Val30Met patients and can be influenced by pre-treatment patient’s features (disease duration, nutritional status, age, severity of neuropathy, and cardiac involvement) [19]. In 2011, tafamidis, an orally administered TTR protein stabilizer, was approved by European Medicine Agency on the basis of its results in study Fx-005 (NCT00409175), a pivotal placebo-controlled study, which enrolled patients with Stage 1 disease and the Val30Met mutation in the TTR gene. Although published data showed that tafamidis is able to delay the course of neuropathy with good preservation of nutritional status, a neurologic progression has been observed in 40–65% of patients after 12 months of tafamidis treatment [20,21,22]. In more recent years, TTR gene silencing therapy with an antisense oligonucleotide (ASO) (inotersen) or a small interfering RNA (siRNA) (patisiran) appeared as a more promising therapeutic strategy for hATTR amyloidosis. TTR silencers provided a real therapeutic revolution, showing evidence that disease progression can be slowed and perhaps reversed [3,4,9,23,24,25].

In particular, in the phase 3, randomized, double-blind, placebo-controlled (2:1), 18-month Apollo study, patisiran has been demonstrated effective in substantially reduce TTR concentration [23], with significant and sustained improvement in polyneuropathy scores [26], autonomic neuropathy [27], quality-of-life profile [28], and some cardiac parameters [29]. For some parameters, such as the modified Neuropathy Impairment Score + 7 (mNIS + 7) and the Norfolk Quality of Life–Diabetic Neuropathy questionnaire total score (Norfolk QOL-DN), a statistically significant improvement was achieved at 18 months compared with placebo, with effects seen as early as 9 months [3]. The interim 12-month analysis of the ongoing lobal open-label extension (OLE) study, which involved patients from the Apollo study and from the 24-month phase 2 single-arm OLE, patisiran appeared to maintain long-term efficacy with an acceptable safety profile [30,31]. Since July 2018, this drug has been available in Italy for patients as compassionate use.

## 2. Patients and Methods

### 2.1. Patients and Outcome Measures

We report here 18 subjects affected by hATTR amyloidosis and treated with patisiran. They were evaluated at the “Regional Centre of Reference for Diagnosis and Treatment of Amyloidosis” of the Unit of Neurology and Neuromuscular Diseases—Department of Clinical and Experimental Medicine—University of Messina (Italy) or at the “Fondazione PoliclinicoUniversitario, A. Gemelli IRCCS,” Unit of Neurology, Largo A Gemelli 8, 00168 Rome, Italy. All 18 subjects underwent neurologic evaluation, each obtaining a Polyneuropathy Disability (PND) score (with higher scores indicating more impaired walking ability), aNeuropathy Impairment Score (NIS) (range, 0 to 192, with higher scores indicating more impairment), and a quality of life assessment with the Norfolk Quality of Life–Diabetic Neuropathy (Norfolk QOL-DN) questionnaire (range, −4 to 136, with higher scores indicating worse quality of life) at baseline and after 6 months of treatment (M6). Fifteen patients were evaluated after 12 months (M12) and 9 patients after 18 months (M18). For these three items, we also recorded retrospective data in all patients at 6, 12, and 18 months before baseline (M-6, M-12, and M-18).

Improvement in the NIS score was defined as a reduction of at least 3 points; the increaseof at least 3 points in this score was considered a sign of clinical worsening. Patients with an NIS score variation between ±2.75 points were considered stable. Similarly, improvement in the Norfolk QOL-DN score was defined as a reduction of at least 4 points; an increaseof at least 4 points in this score was considered a sign of worsening quality of life. Patients with a Norfolk QOL-DN score variation between ±3 points were considered stable.

This study was approved and performed under the ethical guidelines issued by our institutions for clinical studies and was in compliance with the Helsinki Declaration. The study was approved by the ethics committee of our hospitals, and informed written consent was obtained from all the patients(Ethical Committee Code: nr.3/2016, of the 22 march 2016. Name: Ethical Committee of the University Hospital of Messina (address: AOU “G.Martino,” via Consolare Valeria n.1, 98125-Messina (ME), Italy)).

### 2.2. Statistical Analysis

For the purpose of statistical analysis, the patients were divided into sixgroups based on the length of the available observation period before and after treatment, on the presence of a co-treatment with tafamids and on the PND stage at baseline:-Group 1 (all patients) from M-6 to M6.-Group 2 (14 patients) from M-12 to M12.-Group 3 (11 patients) from M-18 to M18.-Group 4 (4 patients with tafamidis as concomitant medication) from M-12 to M12.-Group 5 (8 patients in PND stage 1–2) from M-12 to M12.-Group 6 (6 patients in PND stage 3A and 3B) from M-12 to M12.

The statistical analysis was performed by calculating the mean value and the one sample Wilcoxon test between the deltas of the pre-treatment period and those of the post-treatment period in the sixabove-mentioned groups. Statistical significance was set at *p* < 0.05 (lowered after Bonferroni’s adjustment to <0.008).

## 3. Results

### 3.1. Demographics

The 18 patients harbored six different TTR mutations: Phe64Leu (n.7), Glu89Gln (n.5), Val30Met (n.2), Thr49Ala (n.2), Val122Ile (n.1), and Ala109Ser (n.1) (Table 1). Age at onset varied from 45 to 75 years (mean: 59.6), with hATTR amyloidosis diagnosis set 2.6 years after first symptoms (mean age at diagnosis: 62.2 years). Mean age at first patisiran infusion was 63.8 years. Four patients were also in treatment with tafamidismeglumine; because of poor response, patisiranwas initiatedwithout withdrawing tafamidis, as per the indications of thecompassionate useprogram.

### 3.2. Neurologic Evaluation

At baseline, PND score ranged from 1 to 3b (Table A1). The same result was obtained at last follow up, except for threepatients who improved (two from 3a to 2, and one from 3b to 2) and twopatients who worsened (one from class 2 to 3a and one from class 2 to 3b). The data of the patient who presented the best response to patisiran treatment were previously published [32].

Mean NIS showed a progressive increase in the 18months before starting patisiran, being 70.2 at baseline. This parameter continued to worsen after the first 6 months of treatment (18/18 patients mean NIS at M6: 72) (Figure 1). An improvement started after 12 months (15/18 patients mean NIS at M12: 70.9), persisting after 18 months (11/18 patients mean NIS at M18: 68). Globally, 7/18 (38.8%) patients improved at NIS, 6/18 (33.3%) remained stable, and 5/18 (27.7%) worsened (Table A2).

Similar results were found at NIS lower limb (NIS-LL), a subscale of NIS obtained considering only lower limb motor strength, reflexes, and sensitivity data (range 0–88) (Figure 2). After a continuing increase in the months before, mean NIS-LL was 39.8 at baseline, and worsened after 6 months (18/18 patients mean NIS-LL at M6: 41.4). At the 12-month evaluation, NIS-LL was returned to baseline levels (15/18 patients mean NIS-LL at M12: 39.9) with a considerable improvement registered after 18 months (11/18 patients mean NIS-LL at M18: 37.1). Globally, 6/18 (33.3%) patients improved at NIS-LL, 6/18 (33.3%) remained stable, and 6/18 (33.3%) worsened (Table A3).

The results of the statistical analyses for NIS and NIS-LL are shown in Table 2. To facilitate the comparison of the deltas between groups (with different features and different duration of observation), the values are expressed as mean monthly change. Comparing the pre-treatment deltas vs post treatment deltas, a significant difference was observed in all subgroups except for group 4. Table 2 includes mean monthly change in NIS-LL obtained in the placebo group and in the tafamidis group in the tafamidis trial [33].

### 3.3. Quality of Life Assessment

To assessquality of life, we recorded data from 17/18 patients. Mean Norfolk QOL-DN continued to worsen in the 18 months before patisiran and also after 6 months of treatment (17/17 patients’ mean Norfolk QOL-DN at baseline: 64.2; at M6: 65.7) (Figure 3). A decrease was recorded after 12 months (15/17 patients mean value at M12: 62.7), as well as after 18 months (11/17 patients mean value at M18: 57.4). Globally, 10/17 (58.8%) patients improved at Norfolk QOL-DN, 4/17 (23.5%) remained stable, and 3/17 (17.6%) worsened (Table A4).

The results of the statistical analysis for this scale are shown in Table 3. As for NIS and NIS-LL, no significant difference was found in group4. Instead, a significant difference was observed when comparing the deltas in all the remaining five groups.

### 3.4. Adverse Events

We reported two side effects secondary to the use of premedication drugs: hyperglycemia in one patient (maximum fasting value of 361 mg/dL) and hypertension in two patients (maximum systolic value of 180 mmHg and diastolic of 125 mmHg). These conditions improved after appropriate drug therapy.

Three patients temporarily discontinued the therapy because of hospitalizations, for severe anemia, fever (two distinct episodes), and diarrhea, respectively. All patients started the therapy again after their discharge.

We also reported three deaths. Two patients presented a sudden death, probably of cardiac origin, after 9 and 16 months of therapy, respectively. Another patient died of acomplication aftersevere dehydration.

## 4. Discussion

Real-world, post-marketing, observational studies can provide useful information beyond the confines of traditional clinical trials. This is particularly true when multiple drugs are available and clinicians need to better understand individual indication criteria.

Eighteen hATTR amyloidosis patients, with different genotype and phenotype, were treated with patisiran for a mean period of 14 months. A stabilization of disease progression, with a significant improvement in some cases, expressed by the NIS, NIS-LL, Norfolk QOL-DN, and PND score variation, was observed.

PND score remained stable in 13/18 patients and improved in 3/18. Interestingly, none of the 7/18 patients that were in stage 3a or 3b showed a worsened PND score: four of them remained in the same PND class and three moved to a lower class, expressing a better walking capacity (Table A1). However, in this subgroup of patients, we did not finda significant difference between NIS, NIS-LL, and Norfolk QOL-DN changes before norafter the beginning of treatment (although an analysis showed a *p* < 0.05, whichlost significance after Bonferroni’s adjustment). These results, suggesting the possible presence of a promising efficacy of patisiran even in late stages of disease, need to be further confirmed in larger group of patients. Instead, considering the whole cohort, the analysis of NIS and NIS-LL results showed a global clinical improvement, with a considerable reduction of these scores seen at 18-month follow-up visits (Table 2, Table A2 and Table A3). One limitation of the present study is that only 11/18 patients completed the M18 evaluation, but an improving tendency was already clear at M12 (milestone reached by 15/18 patients) (Table 2). On the other hand, at M6, both mean NIS and NIS-LL monthly changes (+0.31 and +0.27, respectively) suggested a continuation of the progressive clinical worsening registered from M-18 to baseline, although a significant difference was found between the deltas pre- and post-treatment, even in this group of patients (Table 2). In the Apollo study, clinical improvement (represented by change from baseline of mNIS+7) was reported after 9 months of treatment. Our data partially confirm what was shown in the Apollo study, since at 12 months, there wasa reduction in the NIS and NIS-LL values. However, since no reduction in these scores was observed at M6, thoughthere was a statistically significant slowdown in worsening, these results suggest the possible presence of a 6-month latency period prior to benefit of treatment. The samecould be seenfor the patients’ quality of life. Norfolk QOL-DN tended to increase from M-18 to baseline (Table 3, Table A4). The deterioration began to slow significantly in the baseline-M6 period, but we only observed a reduction in the values of this scale in the period M6–M18 (Table 3). Overall, it is not surprising that the worsening of patients’ quality of life goes together with their clinical deterioration in the same period of time. As the same, with the improvement of neurologic condition at M12 and M18, a significant decrease of Norfolk QOL-DN was recorded at the same milestones (Table 3). Finally, no significant differences between pre- and post-treatment results were found in the subgroup of patients also treated with tafamidis. This could be explained by the small number of patients in double treatment, whichcould have made such an analysis inconsistent. Adverse events reported were correlated mainly to the use of premedical drugs. The three deaths reported were considered highly unlikely to be related to patisiran infusion.

## 5. Conclusions

Treatment with patisiran is highly recommended in patients with hATTR amyloidosis. The use of outcome measures such as NIS and Norfolk QOL-DN is useful to monitor treatment response, but it should be taken into consideration that a 6-month latency period could be present before clinical benefits become evident. However, further studies involving a greater number of patients should be necessary to confirm these results.

## Figures and Tables

**Figure 1 brainsci-11-00515-f001:**
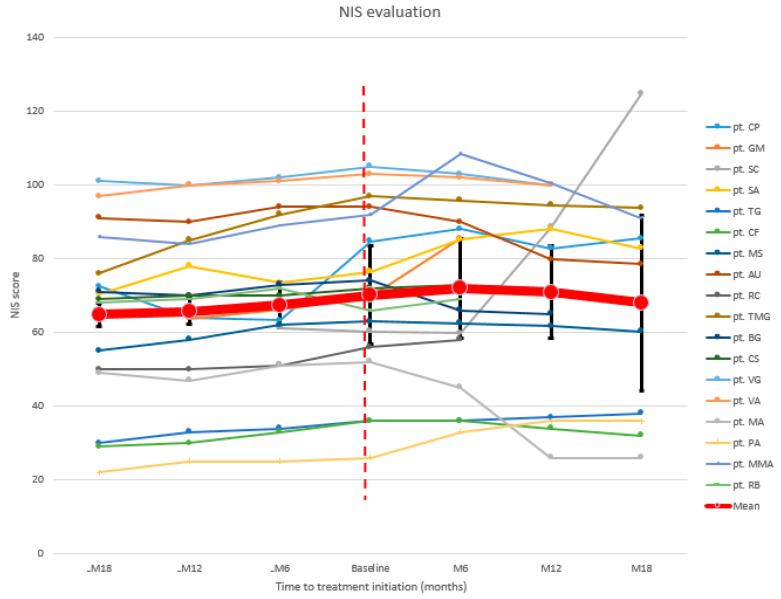
NIS: neuropathy impairment score.

**Figure 2 brainsci-11-00515-f002:**
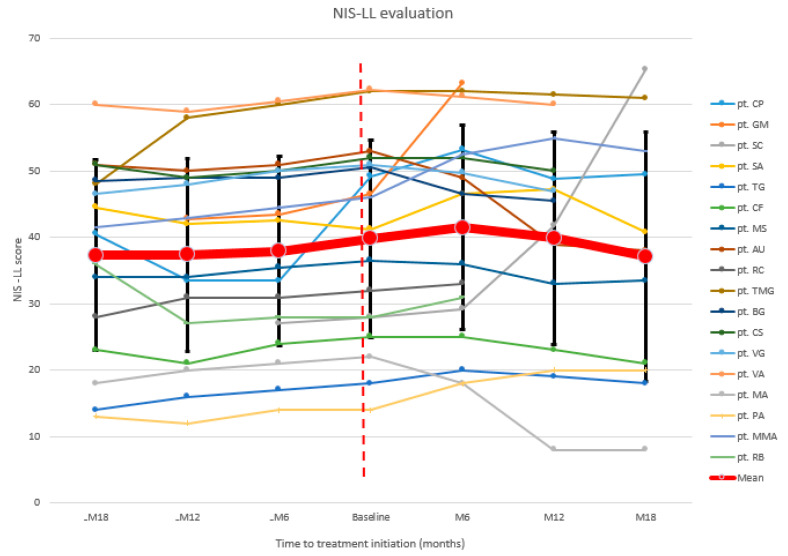
NIS-LL: neuropathy impairment score lower limbs.

**Figure 3 brainsci-11-00515-f003:**
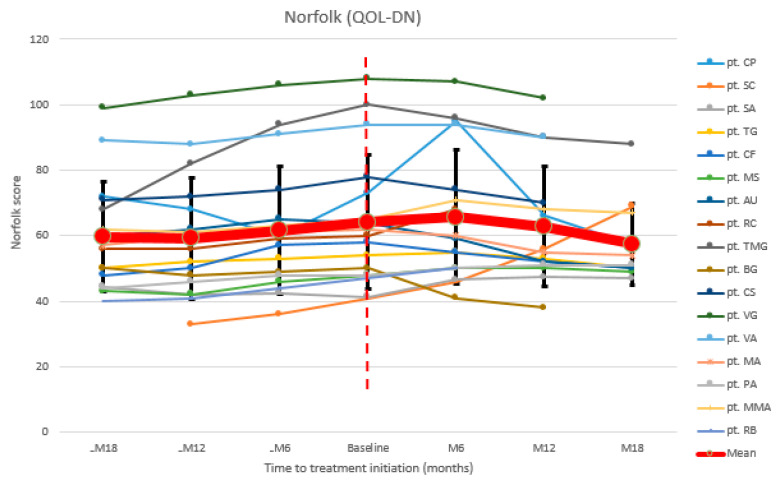
Norfolk QOL-DN: Norfolk quality of life diabetic neuropathy.

**Table 1 brainsci-11-00515-t001:** Demographics, genotypes and concomitant treatments.

Patient	Mutation	Age at Onset	Age at Diagnosis	Age at Baseline	Concomitant Treatments
pt. CP	Phe64Leu	68	72	78	\
pt. GM (d) ^1^	Phe64Leu	63	68	68	\
pt. SC	Ala109Ser	75	71	75	\
pt. SA	Val30Met	59	62	68	\
pt. TG	Phe64Leu	48	54	55	Tafamidis
pt. CF	Glu89Gln	48	54	55	Tafamidis
pt. MS	Phe64Leu	61	66	67	Tafamidis
pt. AU	Phe64Leu	75	77	77	\
pt. RC (d) ^1^	Val122Ile	65	69	70	\
pt. TMG	Glu89Gln	56	57	58	\
pt. BG	Phe64Leu	71	73	74	Tafamidis
pt. CS	Phe64Leu	66	68	69	\
pt. VG	Glu89Gln	48	50	50	\
pt. VA	Val30Met	71	75	75	\
pt. MA	Glu89Gln	45	46	47	\
pt. PA	Thr49Ala	55	56	57	\
pt. MMA	Thr49Ala	46	47	50	\
pt. RB (d) ^1^	Glu89Gln	54	55	57	\

(d) ^1^: deceased.

**Table 2 brainsci-11-00515-t002:** NIS and NS-LL statistical analysis.

Group of Patients	Mean Monthly Change		*p*
	Pre-Treatment	Post-Tratment	
G.1-NIS	0.46	0.31	**0.004**
G.2-NIS	0.43	−0.15	**0.000**
G.3-NIS	0.42	−0.19	**0.002**
G.4-NIS	0.38	−0.23	0.125
G.5-NIS	0.88	−0.24	0.015
G.6-NIS	0.97	−0.67	0.031
G. 1-NIS-LL	0.32	0.27	**0.001**
G. 2-NIS-LL	0.27	−0.11	**0.000**
G. 3-NIS-LL	0.22	−0.13	**0.002**
G. 4-NIS-LL	0.21	−0.20	0.125
G. 5-NIS-LL	0.64	−0.19	0.015
G. 6-NIS-LL	0.49	−0.46	0.031
Placebo group NIS-LL [33]	0.32		
Tafamidis Group NIS-LL [33]	0.16		

G. = Group; G.1 = all patients from M-6 to M6; G.2 = 14 patients from M-12 to M12; G.3 = 10 patients from M-18 to M18; G.4 = 4 patients with tafamidis as concomitant medication from M-12 to M12; G.5 = 8 patients in PND stage 1–2 from M-12 to M12; G.6 = 6 patients in PND stage 3A and 3B from M-12 to M12. In bold, significantvalues.

**Table 3 brainsci-11-00515-t003:** Norfolk QoL statistical analysis.

Group of Patients	Mean Monthly Change		*p*
	Pre-Treatment	Post-Tratment	
G. 1-Norfolk Qol	0.42	0.26	**0.002**
G. 2-Norfolk Qol	0.37	−0.27	**0.001**
G. 3-Norfolk Qol	0.36	−0.28	**0.001**
G. 4-Norfolk Qol	0.38	−0.35	0.125
G. 5-Norfolk Qol	0.63	−0.29	0.023
G. 6-Norfolk Qol	1.03	−1.25	0.031

G. = Group; G.1 = all patients from M-6 to M6; G.2 = 14 patients from M-12 to M12; G.3 = 10 patients from M-18 to M18; G.4 = 4 patients with tafamidis as concomitant medication from M-12 to M12; G.5 = 8 patients in PND stage 1-2 from M-12 to M12; G.6 = 6 patients in PND stage 3A and 3B from M-12 to M12. In bold, significant values.

## Data Availability

The data presented in this study are available on request from the corresponding author.

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
