# Peer review of "Patisiran in hATTR Amyloidosis: Six-Month Latency Period before Efficacy"

_brainsci, 2021, doi:10.3390/brainsci11040515_

Round 1
Reviewer 1 Report
Comments to the Author
This manuscript presents patisiran as a treatment for hereditary amyloidosis associated with mutations in the transthyretin gene (hATTR). The authors administered patisiran to 18 subjects affected by hATTR and examined disease progression up to 18 months post-treatment. The authors found that patisiran displays a 6-month latency period in treatment where deterioration is slowed significantly, and in some cases, reduction of disease progression was observed beyond 6 months of treatment. The paper is well written and organized. However, the presentation of figures and therefore the results are unclear (See comments below). The following are some suggestions for the present manuscript:
- Figures 1, 2, and 3: All three of these figures are presented in the same manner. However, the results appear to contradict the statistical analyses performed claiming a statistically significant difference in mean monthly changes in evaluation score for each group. Especially since the mean change in score for all evaluations is in bold, it seems like there is very little/no change in NIS, NIS-LL, and Norfolk QOL-DN scores. It may be useful to either include supplemental figures or divide these main figures according to which group is being analyzed. This may make the difference in scores more noticeable and seem less contradictory to the statistical analysis performed.
- Discussion: Do the authors speculate on why individual patient outcome varies? Is this something that is potentially mutation-dependent? Or when they start treatment with patisiran?
- Discussion: Please refrain from claiming that patisiran treatment leads to “a significant improvement in the course of the disease.” Instead, I suggest the authors claim that this course of treatment may stabilize disease progression, given the current data presented. The tables in the appendix show that only 3/18 patients displayed an improved NIS score, 4/18 displayed an improved NIS-LL score, and 3/18 patients had an improved QOL-DN score based on the criteria given in the Methods section.
- Appendix A: I suggest including a table like the others in this appendix that shows the PND scores for all patients at all time points. It may also be useful to include a figure like those presented for other evaluations.
Reviewer 2 Report
Thank you for allowing me to comment on the revised version of this manuscript.
The authors added statistical testing and improved the quality of the manuscript by adding a period prior to treatment with patisiran as a form of individual control, which improves the quality of the manuscript. The subgrouping (e.g. G1-G6) is extensive and this reviewer would have preferred a different way of controlling for different follow-up intervals, i.e. least-squares means.
If the present method is applied, I have the following major concerns:
- A test for dependent groups should be performed; i.e. paired t-test on the delta of the means (if the data is indeed normally distributed).
- Adjustment for multiple testing, e.g. bonferroni correction, should be performed.
Minor comments:
- Please reformat the statistical tables so that the different properties of the groups are obvious (e.g. rename groups or provide in legend of the table).
